# Healthcare Received in the Last Months of Life in Portugal: A Systematic Review

**DOI:** 10.3390/healthcare7040122

**Published:** 2019-10-24

**Authors:** Alexandra Pereira, Amélia Ferreira, José Martins

**Affiliations:** 1Community Care Unit of Lousada, Rua de Santo Tirso 70, Meinedo, Lousada, 4620-848 Porto, Portugal; 2Abel Salazar Biomedical Institute, R. Jorge de Viterbo Ferreira 228, 4050-313 Porto, Portugal; 3Medical-Surgical Nursing Department, Nursing School of Coimbra, 3046-841 Coimbra, Portugal

**Keywords:** end-of-life, palliative care, healthcare

## Abstract

*Background*: While evidence demonstrates that end-of-life care practices vary across countries, there is still a lack of evidence regarding the healthcare that is received by adult individuals in the last months of life in Portugal. *Methods*: This is a systematic review that aims to examine the evidence published until 2019 regarding the healthcare received by adult individuals in the last months of life in Portugal. *Results*: Nine studies were included in this review. All of these were quantitative and retrospective studies, used patient records as the source of information, and were conducted in hospital settings. The time frame analysis before death ranged from 48 h to 3 months. The majority of the studies focused on the physical aspects of care, such as physical needs assessment and symptom management, prescribed medication, and invasive interventions. No evidence was found regarding spiritual, psychological, social or cultural needs. *Conclusion*: Although using patient records as a source of information may introduce a bias, this study indicates that there is a greater emphasis on the physical aspects of care and less on the psychological, spiritual or social aspects of care received by adult individuals in their last months of life in Portugal.

## 1. Introduction

Increased life expectancy and better health conditions are allowing people to live longer [1] and to manage many chronic conditions over time. As a result, most people should expect to experience a period of dependency, complex needs, and increased healthcare consumption before dying [2,3].

Chronic diseases are a major cause of death worldwide [4], accounting for 60% of all deaths [5]. The impact of these serious, life-threatening, and life-limiting diseases on society and health systems is enormous. It is estimated that by 2060, 47% of all people will die with serious health-related suffering [6].

In this context, debates about a dignifying death have been ongoing since the 1960s with the start of the hospice movement [7], and end-of-life care is increasingly receiving the attention of the public and policymakers [8]. Thus, palliative care appears to be an ideal approach, providing high-quality care and preventing suffering within the particular setting of the patient [9].

Simultaneously, there is growing discussion and concern about the place of death and people’s preferences. The results of a systematic review that included 210 studies from 33 countries show that the majority of people would prefer to die at home [2]. In Europe, it is known that at least two-thirds of people would prefer to die at home, even though the majority of deaths occur in an institution [10].

Portugal is the 7th most ageing country in the world. By 2050, it will be the second [1]. The above mentioned PRISMA study showed that in Portugal, more than one-half of people would prefer to die at home [10]. Even so, more than 60% of all deaths occur at hospitals/clinics and less than 20% occur at home. Thus, there is a substantial gap between the reality and the population’s preferences for place of death in Portugal [11].

Evidence regarding past trends and projections shows that hospital deaths will continue to increase by more than a quarter until 2030 in Portugal. This is mainly due to the increase in hospital deaths of those aged more than 85 years [12]. Also, Portugal is one of the countries with the highest rate of adults in need of palliative care, and in spite of being considered to have a generalized provision of palliative care [13], there is still an uneven distribution of and access to palliative care [14,15]. The Portuguese provision of palliative care is based in a network model, integrated in the national health system. Palliative care can be provided in three different levels: primary care, hospital care and long term care. Access to palliative care is granted by referral to three types of teams: hospital support teams, palliative care units and palliative care home teams. While remaining below the estimated needs [15], there are more units in the hospital setting than there are teams in the community setting [16].

Although there is evidence from a range of sources that demonstrates that end-of-life care practices vary across countries [17], there is still a lack of evidence regarding the healthcare received by adult individuals in their last months of life in Portugal. As improving care for those at the end of life is quickly becoming an important issue in many countries [18], this systematic review aims to examine the evidence regarding the healthcare received by adult individuals in the last months of life in Portugal.

## 2. Materials and Methods

### 2.1. Search Strategies

This systematic review was undertaken to identify research that characterizes the healthcare received by adult individuals with chronic conditions at the end of life in Portugal. The following databases were researched: CINAHL, Medline, Pubmed, EBSCO, and SCOPUS. A combination of the following terms was used with the Boolean phrase ‘and/or’ in order to maximise the type and range of material captured in the search: ‘end of life’ OR ‘last days of life’ AND ‘care’ AND ‘Portugal’. The search was conducted in English and Portuguese. We also researched a Portuguese research repository called RMP as well as academic electronic repositories to find any missing studies. The ethical procedures were guaranteed through rigorous methodology compliance and respect for the ethical principles that guide health research.

### 2.2. Inclusion and Exclusion Criteria

Publications were selected based on the following inclusion criteria: (1) research that characterized the healthcare received by adult individuals with chronic conditions at the end of life; (2) research conducted in Portugal; (3) research published before 2019. Publications that were based on opinion or commentary, editorials, and conference abstracts were excluded.

### 2.3. Data Extraction

All publications were exported to an Excel database and the duplicates were removed. A data extraction protocol was developed. Results were analysed by two independent researchers to confirm the inclusion criteria. After initial extraction by one of the researchers, the data extracted were cross-checked by the other researcher; consequently, all data were double-checked. When there was disagreement between the two researchers, a third researcher was invited to contribute with an opinion to reach a consensus.

The data extracted from each included publication were as follows: study methods (design, aim, participants, and instruments), study setting, time frame, and outcome measures. All findings were analysed regarding the following aspects: needs assessment and symptom management, prescribed medication, invasive interventions, non-pharmacological interventions, emergency service episodes, recurrent hospitalizations, family care, patients’ choices and preferences, information about diagnostic and/or prognostic, place of death, and after-death care. These aspects were decided a priori and were based on the domains of quality palliative care [19].

### 2.4. Quality Assessment

Study quality was independently assessed by the two researchers using the Risk of Bias Assessment Tool for Non-Randomized Studies (RoBANS) [20]. Criteria included the selection of participants, confounding variables, intervention measurement, blinding of outcome assessment, incomplete outcome data, and selective outcome reporting. Each criterion was evaluated as ‘low risk of bias’, ‘high risk of bias’ or ‘unclear’. In cases of disagreement, each case was discussed with a third researcher.

## 3. Results

### 3.1. Identified Studies

A flow diagram with the selection of cases is detailed in Figure 1 [21]. The review process for the selected studies progressed in three stages: title review, abstract review, and full-text review. Overall, 76 studies were identified through the database search and 6 additional studies were identified through other sources. Five studies were removed as they were duplicates. In the second stage, 60 studies were excluded: 58 did not meet the inclusion criteria and 2 did not correspond to the required format. Seventeen studies were included in the full-text review. Finally, a total of 9 studies that were published during the period under examination were selected for this review.

### 3.2. Study Methods

Detailed information regarding the study methods is available in Table 1. Regarding the study design, we found that eight studies were quantitative descriptive retrospective studies (S1, S2, S3, S4, S5, S7, S8, S9) and one study was a quantitative comparative retrospective study (S6).

In terms of the study aim, five studies aimed to characterize the care provided in a specific hospital service in the last moments of the patients’ lives (S1, S5, S7, S8, S9). One study aimed to evaluate the use that patients made of services provided by a specific hospital in the last moments of the patients’ lives (S2). One study aimed to characterize and compare the health staff records completed in the last moments of the patients’ lives (S6). Two studies aimed to evaluate the use of chemotherapy in the last moments of the patients’ lives (S3, S4).

All studies used patient records as the source of information. Four studies specifically analysed cancer patient records (S1, S2, S3, S5) and one study analysed and compared records of patients admitted to a palliative care unit with records of patients admitted to an internal medicine service (S6). Two studies were exclusively related to palliative care (S1, S5). The sample size ranged from 20 patient records (S5) to 1064 patient records (S4).

Three studies used the Liverpool Care Pathway as the basis for designing the instrument of data collection (S5, S8, S9). All other studies used checklists created especially for that purpose.

### 3.3. Study Setting and Time Frame

Detailed information regarding the study setting and time frame is available in Table 2. All studies were conducted in the hospital setting: two studies were conducted in a palliative care unit (S1, S5), two studies were conducted in an internal medicine service (S7, S8), one study was conducted in a medical oncology department (S3), one study was conducted in a basic emergency service, two studies were conducted in a general hospital (S2, S4), and one study was conducted in an internal medicine service and a palliative care unit simultaneously (S6).

Regarding the time frame, we found no similarities between the studies as the time frame analysis before death defined a priori was very heterogeneous. It ranged from 48 h to 3 months. The majority of the studies had a time frame of 1 month or less (S1, S2, S6, S7, S8, S9). Four studies had a time frame of 1 week or less (S1, S6, S8, S9). Two studies had no time frame analysis before death defined a priori (S4, S5).

### 3.4. Outcome Measures

#### 3.4.1. Needs Assessment and Symptom Management

Physical needs assessment and symptom management were presented in seven studies (S1, S2, S5, S6, S7, S8, S9). In one study, there was no evidence of symptom prevalence. Although the author mentioned the fact that symptoms like pain, agitation, death rattle, nausea or vomiting, and dyspnoea were assessed at least once per shift, there was no record of four-hourly assessments (S5). In all other studies, there is evidence of symptom prevalence. We found 18 different physical symptoms. Pain and dyspnoea were mentioned in seven studies (S1, S2, S5, S6, S7, S8, S9), followed by nausea and vomiting, which were mentioned in six studies (S1, S5, S6, S7, S8, S9). Death rattle was found in five studies (S1, S5, S7, S8, S9) and agitation in four studies (S5, S6, S7, S9). Delirium was mentioned in three studies (S1, S7, S8). Prostration was found in two studies (S7, S9), as was fever (S1, S7). Other symptoms were found once in four different studies: drowsiness, asthenia, and anorexia (S2), obstipation, xerostomia, insomnia and fatigue (S6), confusion (S7), and coughing and haemorrhaging (S1).

The most prevalent symptom was dyspnoea (77.4%) (S8) and the least prevalent was xerostomia (2.1%) (S6). Regarding the prevalence of the three most common symptoms found in the included articles, we found that pain ranged from 22.1% (S7) to 68.6% (S2), dyspnoea ranged from 36% (S1) to 77.4% (S8), and nausea ranged from 3.6% (S9) to 27.1% (S6).

One study described the symptom assessments at 72, 48, and 24 h before death. Dyspnoea, death rattle, and pain prevalence increased as the patient got closer to death, while nausea and vomiting and delirium prevalence decreased as the patient got closer to death (S8).

Three studies also described the absence of recording of any physical symptoms (S2, S6, S9). There was an absence of records for the following symptoms: pain (S2, S6, S9), dyspnoea (S2, S6), nausea and vomiting (S6, S9), agitation (S6, S9), asthenia and anorexia (S2), death rattle (S9), and obstipation, xerostomia, insomnia and fatigue (S6). The absence of records was greatest in the nausea and vomiting assessment (90.4%) (S9).

One study compared the record of physical needs assessment and symptom evaluation between patients who were admitted to a palliative care unit and patients who were admitted to an internal medicine service (S6). There was no absence of records of pain, nausea, and vomiting evaluation in the palliative care unit. All other symptoms had an absence of records in both services, although the prevalence of absence was higher in the internal medicine service. The authors justified this discrepancy by the way in which the needs assessment and symptom evaluation were recorded: free text in the internal medicine service and a systematic flow chart in the palliative care unit.

Although psychological and spiritual needs assessment and symptom management were an outcome measure in one study (S5), the author found no evidence of spiritual or psychological needs assessment in the patients’ records. We found no evidence of social or cultural needs assessment or symptom management in all included studies.

#### 3.4.2. Prescribed Medication

Prescribed medication was mentioned by all studies except one (S4). Regarding the route of administration, the IV route was used in 40% of patients in study S5, 98.8% in study S9, 89.6% in the internal medicine service and 29.2% in the palliative care unit in study S6, and 8.8% in study S2. The oral route was used in 40.4% of patients in study S2 as well as 47.9% in the internal medicine service and 68.8% in the palliative care unit service in study S6. The subcutaneous route was used in 31.6% of patients in study S2, 44.6% in study S9 (for heparin or insulin administration; 13.3% of patients used it for symptomatic control), and 54.2% in the internal medicine service and 79.2% in the palliative care unit in study S6. Study S1 stated that the subcutaneous route of administration was used extensively followed by the oral route but that in most patients, a combination of different routes was used.

Regarding antibiotics, we found their use in six studies: 50% in study S5, 83.9% in study S8, 49.4% in study S9, 77% in the internal medicine service and 47.9% in the palliative care unit in study S6, 9% in study S1, and in 74% of the patients followed by a palliative care team in study S7.

Different types of medicines were found in the included studies. Opioid analgesic usage was found in study S5 (100%), study S2 (41.3%), study S8 (61.3%), study S9 (33.7%), study S7 (54%) and study S1 (85%). The was also evidence of minor opioid (18.8% in the internal service medicine and the palliative care unit) and major opioid usage (20.8% in the internal service medicine and 64.6% in the palliative care unit) in study S6. Non-opioid analgesic usage was found in study S8 (64.5%), in study S9 (37.3%), and in study S1 (18%). Antiemetic usage was found in study S5 (60%), in study S8 (25.8%), in study S6 (31.4% in the internal medicine service and 41.6% in the palliative care unit) and in study S1 (22%). Anticholinergic usage was found in study S5 (50%), study S8 (3.2%), and study S1 (23%). Laxative usage was found in study S1 (26%) and study S6 (52.1% in the internal medicine service and 58.3% in the palliative care unit). Benzodiazepine usage was found in study S8 (9.7%), study S9 (13.3%), and study S1 (33% midazolam and 23% lorazepam). Neuroleptic usage was found in study S9 (19.3%) and study S1 (33% haloperidol). Diuretic usage was found in study S8 (64.5%) and study S1 (6%). Insulin usage was found in study S8 (38.7%) and study S9 (21.7%). Study S8 also described the use of antidiabetics (6.5%), antiarrhythmics (25.8%) and antihypertensives (41.9%). Study S9 also described the use of heparin (32.5%). Study S1 also described the use of corticoids (37% dexamethasone).

Serum therapy was found in four studies: 96.8% in study S8, 98.8% in study S9, and 68.8% in the internal medicine service and 23% in the palliative care unit in study S6. Study S1 stated that 12% of patients were receiving parenteral fluids and that this was usually via subcutaneous hypodermoclysis.

We also found evidence regarding the decision to suspend medication considered non-essential in two studies (S5, S9).

#### 3.4.3. Invasive Interventions

Invasive interventions were described in all studies. Four studies described the prevalence of surgical procedures (S3, S4, S7, S8). Study S3 described that 6% of the patients underwent surgery in the last months of life. The prevalence increased to 65% in study S4. Two studies mentioned central catheter placement: 17% of patients in the last 20 days of life (S7) and 19.4% in the last 72 h (S8).

Three studies found evidence of antineoplastic treatment (S2, S3, S4). Study S4 stated that 86% of their sample underwent antineoplastic treatment, of which 46% underwent radiotherapy and 39% underwent chemotherapy (89% were considered palliative). Radiotherapy was also used in 16% of patients included in study S3 in the last three months of life, mostly for the palliation of cancer-related symptoms. Chemotherapy was used in 5.9% of the patients included in study S2 in the last month of life. In study S3, 66% of patients were treated with palliative systemic chemotherapy in the last three months of life, 37% in the last month and 21% in the last two weeks. These authors also state that 50.2% of patients started a new regimen in the last three months of life and 14.2% in the last month. Of these, 93% were treated with chemotherapy for the first time.

We found evidence regarding complementary diagnosis examinations in five studies (S2, S5, S7, S8, S9). Study S7 stated that 23% of the patients who were followed by a palliative care team made at least one complementary diagnosis examination. Routine blood samples were described in study S5 (40%), study S8 (77.4%), study S9 (54.2%) and study S2 (the sum of examinations done on all the patients included in the sample was 266). Urine samples were described in study S8 (22.6%) and study S2 (the sum of examinations done on all the patients included in the sample was 28). Study S8 also found the following complementary diagnosis examinations performed in the last 72 h of life: gasometry (61.3%), X-ray (45.2%), sputum examination (25.8%), and tomography (6.5%). Study S2 also found the following complementary diagnosis exams performed in the last month of life: x-ray (*n* = 139), tomography (*n* = 18), echography (*n* = 11), endoscopy (*n* = 6), sputum examination (*n* = 5), bronchofibroscopy (*n* = 2), colonoscopy (*n* = 1), and scintigraphy (*n* = 1).

We also found evidence regarding the use of technical instrumental nursing interventions in four studies (S2, S6, S8, S9). Eight different interventions were found: urine catheterization, blood and other derivatives transfusion, vital signs monitoring, blood glucose monitoring, secretions aspiration, placement of peripheral venous access, oxygen therapy management, and placement of feeding tubes. Urine catheterization was found in two studies (56.6% in study S9, 83.3% in the internal medicine service and 14.6% in the palliative care unit in study S6). Blood and other derivatives transfusion was found in two studies (2.4% in study S9 and 12.7% in study S2). Vital signs monitoring was found in three studies (92.8% in study S9, 95.8% in the internal medicine service and 54.2% in the palliative care unit in study S6, and 100% in study S8). Blood glucose monitoring was found in two studies (92.8% in study S9 and 74.2% in study S8). Secretions aspiration was found in two studies (24.1% in study S9 and 61.3% in study S8). The placement of peripheral venous access was found in three studies (95.2% in study S9, 50.0% in the internal medicine service in the last 12 h of life in study S6, and 3.2% in study S8). Oxygen therapy management was found in two studies (8.0% in study S9 and 87.1% in study S8). The placement of feeding tubes was found in two studies: 43.8% in the internal medicine service in study S6 and 48.4% in study S8, of which 9.7% were placed in the last 72 h and only 12.9% were recorded as not intubating the patient.

We also found the decision to suspend invasive interventions in two studies (S9, S5). In study S9, medication considered non-essential was suspended in 14.5% of patients. This decision was also recorded in 95% of the patients included in study S5.

#### 3.4.4. Non-Pharmacological Interventions

Non-pharmacological interventions were present in three studies (S5, S8, S9). We found five different non-pharmacological interventions described: hygiene care, oral care, comfort care, patient positioning, and incontinence management. Incontinence management was recorded in all three of these studies. In study S5 and study S8, there was evidence of this care in all patients’ records. Oral care was found in two studies (S5, S9). There was no evidence of this care in any of the analysed cases in study S5 and it was recorded in 2.4% of the cases in study S9. Studies S8 and S9 mentioned patient positioning (100% and 65.1%, respectively) and comfort care (58.1% and 62.7%, respectively). Hygiene care (43.4%) was described only in study S9.

Study S9 also described the absence of a record of non-pharmacological interventions: 54.2% hygiene care, 97.6% oral care, 37.3% comfort care, 34.9% patient positioning, and 9.6% incontinence management.

#### 3.4.5. Emergency Service Episodes

Emergency service episodes were mentioned in two studies (S2, S3). Although one of the included studies was conducted in the basic emergency service (S9) regarding the care received by patients who died there, we found no information regarding different emergency service episodes and so this study was not considered in this outcome measure.

In study S2, 60.7% of patients had at least one emergency room visit in the last month of life. The mean was 1.1 emergency room visits. In study S3, 80% of patients had at least one emergency room visit in the last three months. The mean was two emergency room visits. Of those, 15% were due to chemotherapy treatment-related toxicity.

#### 3.4.6. Recurrent Hospitalizations

Recurrent hospitalizations were mentioned in four studies (S2, S3, S4, S7). In study S2, 84.7% of patients had at least one hospital admission in the last month of life. The mean was 10.6 days of admission. In study S7, 30% of patients had at least one hospital admission in the last year of life. Of those, 7% had more than one hospital admission. The mean was 14.3 days of admission. In study S3, 96% of patients had at least one hospital admission in the last three months of life. Of those, 16% were due to chemotherapy treatment related toxicity and 2.4% were admitted to the intensive-care unit. The mean was 16 days of admission. In study S4, 34% of patients were admitted to the palliative care unit during the period of analyses.

#### 3.4.7. Family Care

The record of family care was not consistent in any of the included studies. We found evidence of family involvement in two studies regarding permission for the presence of the patients’ families (S6, S9). In study S6, there was a recording of this aspect in 16.6% of the patients admitted to the internal medicine service and in 25.8% of the patients admitted to the palliative care unit. In study S9, this aspect was recorded in 48.2% of the cases.

#### 3.4.8. Patient’s Choice and Preferences

Patient’s choice and preferences were mentioned in four studies (S5, S6, S8, S9). Study S6 mentioned that several patients’ choices and preferences were recorded even though the authors did not describe them. The recording was higher in the palliative care unit (66.7% as opposed to 4.2% in the internal medicine service).

Study S9 stated that there was no record of families’ or patients’ expressed choices or preferences regarding the order to not resuscitate. Two studies stated that there was a recording of a do not resuscitate order (100% in S5 and 58.1% in S8), although it is not clear whether this decision was made with the patients’ or families’ consent.

On the other hand, we found evidence regarding the use of cardiopulmonary resuscitation manoeuvres in three studies: 6.5% in study S8, 26.3% in study S9, and 21% in study S7. The last of these also stated that no manoeuvres were made in the patients that were followed by a palliative care team.

#### 3.4.9. Information about Diagnostic and Prognostic Status

Information about diagnostic and/or prognostic status was present in four studies (S5, S6, S7, S9). The confirmation of the contact details of family members was described in study S9 (39.8%) and in study S5 (80%). Information given to the family about the patient’s diagnostic and clinical status was present in study S5 (95%), study S9 (30.1%), and study S6 (16.6% in the internal medicine service and 25.8% in the palliative care unit). The healthcare plan was discussed with the family in study S5 (75%) and study S9 (20.5%). In study S7, there was no evidence of healthcare discussion with the palliative care patients’ families in 26% of the cases. Only in study S5 was there evidence of healthcare discussion with the patient (65%) and of patients’ knowledge about their diagnostic status (70%). Information regarding imminent death given to the family was found in study S5 (85%) and study S9 (30.1%).

#### 3.4.10. Place of Death

Place of death was described in eight studies (S1, S2, S4, S5, S6, S7, S8, S9). In six studies, the place of death was considered an inclusion criterion; as a consequence, all patients in those studies died in the place where the study was conducted. All patients died in the internal medicine service in two studies (S7, S8). The place of death was the palliative care unit for all patients in two studies (S1, S5). The basic emergency room was the place of death for all patients in one study (S9). In one study, half of the patients died in the internal medicine service and half of the patients died in the palliative care unit (S6).

In two studies, the place of death was not considered an inclusion criterion (S2, S4). In study S4, the place of death was the palliative care unit in 22% of cases. Another service in the hospital was the place of death in 22% of cases. Five percent of the patients died at home, 2% died in another hospital and 1% in another place. The place of death was unknown in 49% of cases. The authors presumed that the patients died at home, although they could not find evidence (S4). In study S2, the place of death was a service of the public hospital in 87% of cases, emergency room in 6%, home in 3%, home residence in 2%, and private hospital in 0.4%.

#### 3.4.11. After-Death Care

After-death care was mentioned in two studies (S5, S9). We found that these studies considered after-death care recorded in the patients’ files to be the care provided to the deceased’s body, information given to the family, and administrative procedures regarding documentation. In one study (S9), care provided to the deceased’s body was recorded in 16.7% of the analysed cases, information given to the family was recorded in 10%, and administrative procedures regarding documentation in 8.8%. In the other study (S5), the prevalence of records was higher (85%, 85% and 75%, respectively), perhaps due to the fact that it was conducted in a palliative care unit. This study also described that there was no evidence that bereavement care was offered to the family.

### 3.5. Quality of Studies

The quality of the included studies is summarized in Table 3. Regarding the risk of bias as a result of participant selection, only one study was evaluated as ‘unclear’ (S8). All studies were evaluated as having a ‘high risk of bias’ for confounding variables and as a ‘low risk of bias’ for the blinding of outcome assessment. In appraising the risk of bias as a result of inappropriate intervention measurement, four studies where evaluated as having a ‘high risk of bias’ (S1, S4, S7, S8) and one as ‘unclear’ (S1). In terms of the incomplete outcome data, five studies were evaluated as ‘unclear’ (S1, S2, S3, S5, S6) and one as having a ‘high risk of bias’ (S7). The selective outcome reporting was evaluated as ‘unclear’ in three studies (S1, S4, S8) and as having a ‘high risk of bias’ in one study (S6). Most of the criteria were classified as ‘unclear’ or having a ‘high risk of bias’ in two studies (S1, S8).

## 4. Discussion

Although the majority of palliative care research in Portugal is done to improve individual-level practice [31], there still is a lack of evidence regarding the healthcare received by adult individuals in the last months of life in this country. This review included eight quantitative descriptive retrospective studies and one quantitative comparative retrospective study. All of these used patient records and had small samples, with most ranging from 20 to 300 patient records. Only one study had a sample of more than 1000 participants [25]. Different instruments were used in the data collection. All studies used own-designed checklists and three of them used the Liverpool Care Pathway as a base. In fact, in the palliative care field it is difficult to achieve adequate sample sizes and due to ethical concerns, patient records are usually an acceptable source of information. In the future, research in palliative care needs to use standardized data collection tools and registries that will permit the development of an evidence-based medicine culture in this context [32]. In spite of recent controversy and consequent disuse of the Liverpool Care Pathway, mainly due to poor leadership and inadequate training of staff, towards the end of the last decade, it was widely welcomed and hugely supported by many health centres and was part of many countries’ governments’ strategy. Hence, it is understandable that those studies used an international recommendation as the basis for creating their own instrument of data collection [33].

All included studies were developed in the hospital setting. Three studies were totally or partially developed in palliative care units. Even though adult individuals with chronic diseases spend most of their time at home, in Portugal they usually die in hospital [11]. Also, most palliative care research is conducted during an academic course and as time can be a concern, researchers tend to choose contexts where data collection can be easier. The development of research in non-hospital settings is one of the major gaps in palliative care investigations in Portugal [31]. The care provided at the end of life has become an important issue in many countries [8,18]. In Portugal, palliative care is a relatively recent phenomenon: The first unit emerged in 1992 and the first community-based team appeared in 1996. Specific legislation and a national development strategic plan were only published in the last 10 years [14]. Thus, there has been a growing interest in this area and it is expected that some studies regarding end-of-life care would be conducted specifically in this context.

The time frame of analysis before death was very heterogeneous among the included studies. The majority had a time frame of 1 month or less and four of those were one week or less. Two studies had no defined time frame analysis. This is consistent with international follow-back surveys [8,18]. This result may, in part, explain the results we obtained in the outcome measures, as the most studied measures we found in the included studies are related to the physical aspects of care like the place of death, prescribed medication, needs assessment and symptom management, and invasive interventions.

Physical needs assessment and symptom management were described in seven studies and in six of those, there was evidence of symptom prevalence. Pain and dyspnoea were the most assessed physical symptoms. Dyspnoea had the highest prevalence of any physical symptom and xerostomia has the lowest. Only one study had psychological and spiritual needs assessment and symptom management as an outcome measure, even though the author found no evidence of their assessment. No evidence was found regarding social and cultural needs assessment and symptom management. Even so, in some studies the authors described an absence of a recording of symptoms, and the fact that most contexts did not use a systematic evaluation chart may have also influenced these results.

Prescribed medication was mentioned in eight studies. Regarding the route of administration, there seems to be a predominance of intravenous (IV) route usage in the internal medicine services and subcutaneous route usage in the palliative care units. The oral route usage rate may be influenced by the time frame analysis as it is expected to decrease as the time of death comes nearer. Antibiotic prescription was found in six studies. We found high percentages of prescription in all settings: the highest percentage was 83.9% [29] and the lowest percentage was 9% [30]. Antibiotic prescription at the end of life has raised some controversy, and thus patients in the final stage of an advanced illness often face challenging decisions about medical care. The prescription of antibiotics is one of these as infections and febrile episodes are among the most common acute complications experienced by these patients [34].

There seemed to be a general concern regarding analgesic usage. Opioid analgesic usage was detailed in seven studies and non-opioid analgesic usage was mentioned in three studies. We believe this is related to the fact that pain is being recognized as a common symptom affecting people at the end of life. It is also recognized that pain assessment and management practices undertaken by healthcare professionals have direct implications for pain control [35]. It also seems that the usage of symptom assessment scales can improve the early detection and management of symptoms.

Invasive interventions were described in all studies. Surgical procedures were described in four studies, antineoplastic treatments in three studies, complementary diagnosis examinations in five studies, and technical instrumental nursing interventions in four studies. There is growing controversy regarding the use of invasive interventions at the end of life, although some authors regard invasive treatments as supportive measures rather than curative [36]. Even so, in palliative care, symptom management and the decisions made by health professionals regarding treatment have repercussions for the comfort of the patient. Frequently, the identification of the proximity of death represents an inflection point to avoid the unnecessary suffering for the patient [37]. However, recognizing a patient’s proximity to death is difficult and complex [38,39]. Thus, although we believe that the suspension of invasive interventions should be a general practice, it was only found in two studies.

There seemed to be less attention given to non-pharmacological interventions as they were described in only three studies, which mentioned hygiene care, oral care, comfort care, patient positioning, and incontinence management. This might be explained by the fact that health professionals usually do more activities than those they record, especially during the day when numerous activities are performed. Evidence also shows that health professionals prioritize the recording of physical symptom assessment and technical interventions rather than basic needs assessment or educational activities [40].

Information about diagnostic and prognostic status and patients’ choices and preferences were described in four studies. This is consistent with other studies, where the focus of health professionals is allocated towards specific interventions at the end of life, rather than improving communication skills coupled with health system changes needing to be implemented [41].

Recurrent hospitalizations were described in four studies and emergency service episodes were described in two studies. Although we found no information regarding the care received while the patients were at home, the prevalence found in both cases might indicate that this was not adequate or sufficient. The use of outpatient palliative care is usually a recommendation to reduce emergency service usage and recurrent hospitalizations [42].

Family care was described in two studies. Even so, these studies only mentioned permission for the presence of patients’ families. Important aspects related to family care, such as emotional support, promotion of shared decision-making, and attention to the spiritual needs of the family, were not described in any of the included studies.

After-death care was described in two studies, regarding records related to the care provided to the body of the deceased, information given to the family, and administrative procedures regarding documentation. We found no evidence in relation to grief and bereavement counselling. This might be considered a surprising result as some authors state conceptually and practically that grief intimately relates to palliative care and end of life, as both domains regard the phenomena of loss and suffering [43].

## 5. Conclusions

This study provided baseline evidence of the healthcare received by adult individuals with chronic disease in the last months of life in Portugal. All included studies used patient records as the source of information. Most studies had small samples that characterized the healthcare received in a specific hospital or hospital service. No evidence was found regarding the care received in a community setting, homecare services or long-term facilities. There also was a very heterogeneous time frame analysis before death.

The majority of the included studies emphasized in their analyses the place of death, prescribed medication, and needs assessment and symptom management. Also, greater importance was given to physical needs assessment and symptom management. Psychological and spiritual needs had less representativity and there was a total absence of social and cultural needs. We also found that patients’ choices and preferences, information about diagnostic and prognostic status, recurrent hospitalizations, and invasive interventions were analysed in almost half the studies. There was less evidence regarding non-pharmacological interventions, after-death care, family care, and emergency service episodes. This study also shows that end of life care may vary due to the place where it is provided and also due to the practitioner’s specialty.

The diversity of outcome measures and different instruments used in the data collection in all studies included made it difficult to compare the results and draw definite conclusions. The limitations of this review depend to a large extent on the quality of the studies that were included as we detected a high risk of bias during the quality assessment. Although we included peer-reviewed papers in English and Portuguese, the poor quality of some abstracts might be considered a limitation during the data extraction. Also, the fact that the studies included used patients records as source of information might be considered a bias, as some interventions may not be documented by all professionals. Despite these limitations, we consider that this systematic review is a valuable contribution as it is the first review to focus on the healthcare received in the last months of life in Portugal.

We suggest that domains of quality palliative care are used to define outcome measures in similar studies so that it is possible to obtain more comparable results in future research. Also, research based on patient, caregiver, and family opinions regarding end-of-life care in various settings of care should be developed in the future.

## Figures and Tables

**Figure 1 healthcare-07-00122-f001:**
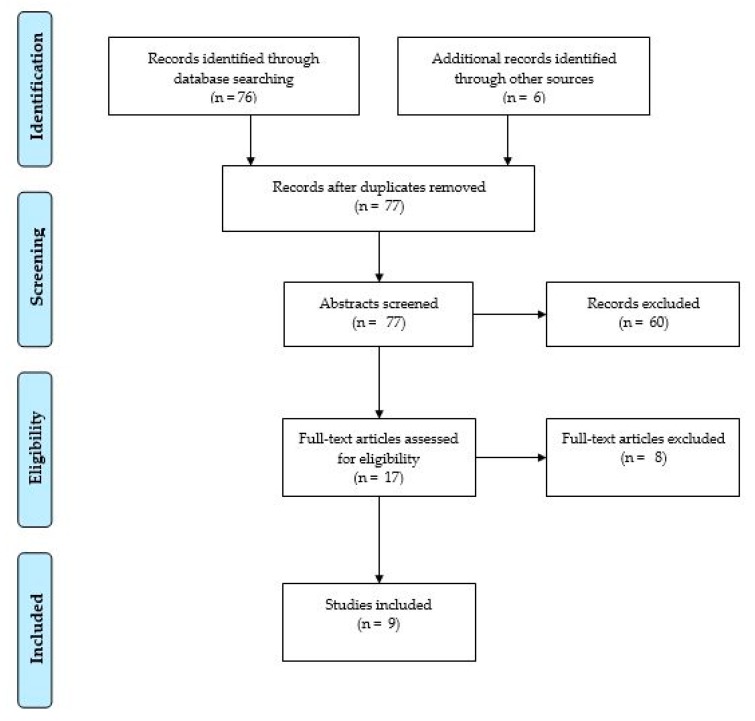
PRISMA flowchart of the review phases, adapted from Moher et al. [21].

**Table 1 healthcare-07-00122-t001:** Included studies: study design, aim, participants and instruments.

Article Number	Author	Study Design	Aim	Participants	Instruments
S1	Gonçalves et al. (2003) [22]	Quantitative retrospective study	To audit experience in the care of patients in the last 48 h of life	300 palliative care cancer patient records	Own-designed checklist
S2	Feio (2006) [23]	Quantitative descriptive retrospective study	To evaluate the use that patients with advanced cancer made of the hospital services	118 cancer patient records	Own-designed checklist
S3	Braga et al. (2007) [24]	Quantitative retrospective study	To evaluate the use of chemotherapy in the last 3 days of life in adult patients with advanced solid tumors	319 cancer patient records	Own-designed checklist
S4	Gonçalves & Goyanes (2008) [25]	Quantitative descriptive retrospective study	To determine the proximity of chemotherapy use to the patient’s death	1064 cancer patient records	Own-designed checklist
S5	Silva (2009) [26]	Quantitative descriptive retrospective study	To characterize how patients are cared for in the last days of life in a palliative care unit	20 palliative care cancer patient records	Checklist based on the Liverpool Care Pathway
S6	Carneiro et al. (2009) [27]	Quantitative comparative retrospective study	To characterize and compare the record of health problems, intervention plans and agonic phase detection in two different hospital services	96 patient records (48 admitted to an internal medicine service and 48 admitted to a palliative care unit)	Own-designed checklist
S7	Pulido et al. (2010) [28]	Quantitative retrospective study	To characterize the care provided to patients in an internal medical service	285 patient records	Own-designed checklist
S8	Delgado (2012) [29]	Quantitative descriptive retrospective study	To describe the healthcare provided to terminal patients in the last 72 h of life in an internal medicine service	31 patient records	Checklist based on the Liverpool Care Pathway
S9	Pereira et al. (2017) [30]	Quantitative descriptive retrospective study	To describe the care provided by the nursing staff of the basic emergency service to end-of-life patients	83 patient records	Checklist based on the Liverpool Care Pathway

**Table 2 healthcare-07-00122-t002:** Included studies: study setting and time frame.

Article Number	Author	Study Setting	Time Frame
S1	Gonçalves et al. (2003) [22]	Hospital setting Palliative care unit	48 h before death
S2	Feio (2006) [23]	Hospital setting	1 month before death
S3	Braga et al. (2007) [24]	Hospital setting Medical oncology department	3 months before death
S4	Gonçalves & Goyanes (2008) [25]	Hospital setting	Not defined
S5	Silva (2009) [26]	Hospital setting Palliative care unit	Not defined
S6	Carneiro et al. (2009) [27]	Hospital setting Internal medicine service Palliative care unit	5 days before death
S7	Pulido et al. (2010) [28]	Hospital setting Internal medicine service	20 days before death
S8	Delgado (2012) [29]	Hospital setting Internal medicine service	72 h before death
S9	Pereira et al. (2017) [30]	Hospital setting Basic emergency service	1 week before death

**Table 3 healthcare-07-00122-t003:** Quality of studies.

Risk of Bias	S1	S2	S3	S4	S5	S6	S7	S8	S9
Selection of participants	○	○	○	○	○	○	○	X	○
Confounding variables	□	□	□	□	□	□	□	□	□
Intervention measurement	□	○	X	□	○	○	□	□	○
Blinding of outcome assessment	○	○	○	○	○	○	○	○	○
Incomplete outcome data	X	X	X	○	X	X	□	○	○
Selective outcome reporting	X	○	○	X	○	□	○	X	○

○: Low risk of bias □: High risk of bias X: Unclear.

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
