# Peer review of "Healthcare Received in the Last Months of Life in Portugal: A Systematic Review"

_healthcare, 2019, doi:10.3390/healthcare7040122_

Round 1

Reviewer 1 Report

line 102: Probably there is a mistake. In the Figure1, 58 studies were excluded. It is necessary to check this numbers on the line 102. Perhaps it could be 58 and 56 instead of 55 and 53

line 110: "we found that seven studies were quantitative descriptive retrospective studies (S1, S2, S3, S4, S5, S7, S8, S9)". There are eight studies.  

Author Response

Dear reviewer,

Thank you for your suggestions.

Here’s the point to point response:

Line 102: There was a mistake. The second reviewer suggested an update of the research until 2019. We found two more articles, but they did not meet the inclusion criteria, and so, no relevant changes were made to the review. As we did that, we corrected the numbers accordingly.

Line 110: There was a mistake. It was corrected. There are eight studies.

Thank you once again.

Best regards.

Alexandra Pereira

Reviewer 2 Report

This paper is a systematic review of the healthcare patients receive at the end of their lives in Portugal. The authors reviewed publications through 2017 in English and Portuguese using PRISMA guidelines and include a review of the strength of evidence. They found 9 articles to include that showed that physical needs and symptom management were the primary outcomes managed and that prescription medications and invasive procedures were the common interventions. This is an important study to add to the literature in order to better understand healthcare practices in Portugal at the end of life.

1. It would be helpful for the authors to define how palliative care is practiced in Portugal for readers unfamiliar with the Portuguese healthcare system For example, in the US, patients receiving end of life care may be receiving care from palliative care specialists or hospice specialists in the last 6 months of life. The management of care between these two specialties are similar, but not the same.

2. If possible, the authors should update their search through 2019. If this is not possible, they should explain why they chose 2017 as a cut off year.

3. Much of the care provided for patients at the end of life varies by the practitioner’s specialty. Palliative care and hospice specialists have expertise on providing care at this phase of life so the interventions used by them compared to non-hospice and palliative practitioners may vary because of this reason.

4. The non-pharmacological interventions described are often not documented in the patient charts in the US by healthcare physicians. While the authors explain in the discussion, this may be due to health professionals not documenting actions they perform, they should consider whether these are actions that differ by the type of health professional.

Author Response

Dear reviewer:

Thank you for your suggestions.

Here’s the point to point response:

1. We also added a paragraph to the introduction section regarding palliative care provision in Portugal, so that the readers can become aware of the Portuguese health care system;2. We updated the search until 2019. We found two more articles, but they did not meet the inclusion criteria, and so, no relevant changes were made to the review. 3 . We also updated our conclusions, reinforcing that end of life care may vary due to the place where it is provided and also due to the practitioner’s specialty. 4. We also updated our conclusions, reinforcing that non documentation on the included studies may be a bias.

Thank you once again.

Best regards.

Alexandra Pereira